# Odor Identification Ability as a Mediator of Schizotypal Traits and Odor Hedonic Capacity in Non-Clinical Children and Adolescents

**DOI:** 10.3390/brainsci12050534

**Published:** 2022-04-22

**Authors:** Ying Chen, Yuyang Zhan, Yiqi Qiu, Jiubo Zhao, Laiquan Zou

**Affiliations:** 1Chemical Senses and Mental Health Lab, Department of Psychology, School of Public Health, Southern Medical University, Guangzhou 510515, China; 3197010017@i.smu.edu.cn (Y.C.); zhanyy0704@outlook.com (Y.Z.); yiqiqiu0825@foxmail.com (Y.Q.); zhjb@smu.edu.cn (J.Z.); 2Department of Psychiatry, Zhujiang Hospital, Southern Medical University, Guangzhou 510280, China

**Keywords:** schizotypal traits, odor identification ability, olfactory hedonic capacity, children and adolescents

## Abstract

Background: Previous studies have investigated the relationship between schizotypal traits and odor identification ability as well as the relationship between schizotypal traits and odor hedonic capacity in adults. However, very little is known about the relationship among these three factors, especially in children and adolescents. The current study aimed to explore the relationship among these three factors in children and adolescents as well as the potential role of odor identification ability. Method: A total of 355 non-clinical children and adolescents (aged 9–16 years) were recruited in the study. They were asked to complete the Universal Sniff Test (U-Sniff), the Chemosensory Pleasure Scale for Children (CPS-C), and the Schizotypal Personality Questionnaire—Child (SPQ-C). Results: The SPQ-C scores were negatively correlated with both the odor identification scores and the odor hedonic scores (*p* = 0.022 and *p* < 0.001, respectively). Only the interpersonal–affective factor of the SPQ-C was negatively correlated with the odor identification scores (*p* = 0.031). The odor identification scores were significantly positively associated with the odor hedonic scores (*p* < 0.001). Moreover, the relationship between odor hedonic capacity and schizotypal traits, especially the interpersonal–affective factor, was mediated by odor identification ability. Conclusions: Schizotypal traits were negatively correlated with both odor identification ability and odor hedonic capacity in children and adolescents, while odor identification ability was found to mediate the relationship between odor hedonic capacity and schizotypal traits, especially the interpersonal–affective factor. Our study indicated that improving odor identification ability through olfactory training may have a positive influence on odor hedonic capacity in individuals with schizotypal traits.

## 1. Introduction

Anhedonia, the declining capacity to experience pleasant emotions, is a marker of mental health disorders, including schizophrenia [1,2] and depressive disorders [3,4]. Compared with other neurobehavioral probes of hedonic capacity, olfactory measures may be more sensitive because there are direct and mutual connections between olfaction and brain regions that modulate olfactory and emotional processing [5]. Previous studies have suggested that schizophrenia patients experience abnormal hedonic processing of odor [5,6,7].

Schizotypal traits are multimodal entities that exist on a psychotic continuum and are risk factors for schizophrenia-spectrum illnesses and full-blown psychosis [8,9,10]. Compared with healthy controls, individuals with high schizotypal traits are more likely to suffer anhedonia, including social and physical anhedonia, than those with low traits [11,12,13]. It has been observed that individuals with schizotypal traits present more severe olfactory anhedonia than healthy controls [14]. Individuals with schizotypal traits also showed an attenuated form of odor identification impairment but still outperformed schizophrenia patients [15,16,17]. In addition, odor identification ability was closely related to anhedonia in individuals with schizotypal traits [13,17].

Although much research has been carried out on the relationship between schizotypal traits and odor identification and the relationship between schizotypal traits and odor hedonic capacity, the role of odor identification ability in the relationship between schizotypal traits and odor hedonic capacity remains unclear. Moreover, studies that examine individuals with schizotypal traits are mostly focused on adults [8,14,18], while those that center on children and adolescents are rare. Much uncertainty still exists about the relationship between schizotypal traits and olfactory function in children and adolescents.

The present study aimed to explore the relationship between schizotypal traits, odor identification ability, and odor hedonic capacity in children and adolescents. We hypothesized that (1) schizotypal traits are negatively correlated with both odor identification ability and odor hedonic capacity in children and adolescents and (2) odor identification ability mediates the relationship between schizotypal traits and odor hedonic capacity.

## 2. Materials and Methods

### 2.1. Participants

Four hundred and sixty-eight children and adolescents aged 9–16 years old agreed to participate in this study during the period 1 January to 31 March 2020. All the participants were from local primary and middle schools in Guangdong, China. The exclusion criteria were as follows: (1) had suffered a cold and upper respiratory infections in the past week; (2) had a history of nasal diseases, brain injury, or neuropsychiatric disorders; (3) took psychotropic medications. After excluding 61 participants, 407 participants finished the tests. We then excluded 52 participants who lied on the questionnaire. Finally, 355 participants aged 9–16 (147 boys and 208 girls, mean age = 12.38 ± 3.48 years) remained for recruitment in the sample (see Figure 1). As this was a cross-sectional study, a 0.95 power estimate, an effect size of 0.3, and an α of 0.05 were used in the G*Power program to calculate the necessary sample size [19]. The sample size in our study was far larger than the result proposed by the G*Power program. All the testing processes were conducted in quiet and well-ventilated rooms. This study was approved by the Ethics Committee of Southern Medical University in Guangzhou, Guangdong, China. Each participant was offered a description of the study before providing informed consent. The study was conducted with the informed consent of the legal guardians of the children and adolescents.

### 2.2. Measures

#### 2.2.1. The Universal Sniff Test (U-Sniff) 

Odor identification ability and odor hedonic capacity were tested using the U-Sniff test [20]. Before presenting the odor in the test, four descriptors (pictures and words) were read and shown to the children and adolescents. They were asked to choose between four descriptors after the odor’s presentation. The correct choices were given a score that ranged from 0 to 12 points. The higher the score, the better the odor identification ability. Participants were also required to evaluate the odor’s pleasantness subjectively from 1 (“very much dislike this odor”) to 9 (“very much like this odor”). All of the ratings were summed to create an overall score. The higher the overall score, the better the odor hedonic capacity. The U-Sniff test had a test–retest reliability of 0.71 in eight countries, which represents a good level of reliability and validity [21,22,23].

#### 2.2.2. Chemosensory Pleasure Scale for Children (CPS-C) 

Odor hedonic capacity was also tested using the CPS-C [24,25]. The CPS-C is a 12-item questionnaire graded on a 6-point Likert scale and consists of three dimensions (nature, food, and imagination) that correspond to the hedonic capacity of eating, anticipating food, and smelling natural scents, respectively. Participants were required to respond with a score ranging from 1 (“not at all true for me”) to 6 (“very true for me”). The higher the score, the better the odor hedonic capacity. The test–retest reliability of the CPS-C was 0.72 [24].

#### 2.2.3. Schizotypal Personality Questionnaire—Child (SPQ-C)

Schizotypal traits in children and adolescents were tested using the SPQ-C [8,26]. The SPQ-C is a 22-item yes/no inventory (1 = Yes, 0 = No) that consists of three main factors of schizotypal traits: perceptual and speech deficits, odd behavior, and the interpersonal–affective factor. The relevant items were added together to calculate the subscale scores and the total score for the whole scale. The higher the score, the higher the schizotypal trait. The test–retest reliability of the SPQ-C over 3 months, 6 months, and 12 months was 0.67, 0.64, and 0.55, respectively [8].

### 2.3. Data Analysis

Data analysis was completed with SPSS 26.0 and AMOS 26.0 (SPSS Inc., Chicago, IL, USA). All statistical tests were 2-sided, and an *α* = 0.05 was considered to be significant. We first checked the normality of the distribution in all datasets, and all of the variables were normally distributed except for odor identification. Aside from the association between odor identification and these variables, which was examined using Spearman’s correlation analysis, the associations between other components were examined using Pearson’s correlations. A partial correlation analysis was also employed to control for the influence of age and gender. Mediating analysis was conducted to explore the role of odor identification in the relationship between schizotypal traits, especially the interpersonal–affective factor of the SPQ-C, and odor hedonic capacity measured by the U-Sniff and the CPS-C. Finally, chi-square difference tests were performed to examine the gender invariance of children and adolescents in the mediating models. The bootstrap mediation technique was continuously applied to 5000 samples with replacement to determine the statistical significance of the mediating effects.

## 3. Results

### 3.1. Correlation Analyses

The SPQ-C score was significantly negatively correlated with the odor identification scores, odor pleasantness scores, total CPS-C scores, and CPS-C subscale scores, including CPS-C nature and CPS-C imagination (all *p* < 0.05) but not CPS-C food (*p* = 0.081) (see Table 1). Only the interpersonal–affective factor of SPQ-C was significantly negatively correlated with odor identification scores (*p* < 0.05). Odor identification scores were significantly positively associated with odor pleasantness, total CPS-C scores, and all of the subscale scores in the CPS-C (nature, food, and imagination) (all *p* < 0.05). Odor pleasantness had a positive association with the total CPS-C scores, including all of the factors in the CPS-C (nature, food, and imagination) (all *p* < 0.05). After performing a partial correlation analysis for potential confounders such as age and gender, the associations remained significant (all *p* < 0.05) excluding the correlation between odor identification and CPS-C imagination (*r* = 0.08, *p* = 0.132).

### 3.2. Mediation Analysis 

#### 3.2.1. The Mediating Effect of Odor Identification on Schizotypal Traits and Odor Hedonic Capacity Assessed by U-Sniff

We treated the total SPQ-C score as the independent variable, the odor pleasantness score as the dependent variable, and the odor identification score as the mediating variable (see Figure 2a). Based on the results of the correlation analysis, a mediating model might exist. The mediating effect of odor identification was tested by using the bias-corrected bootstrap method. In Model 1, zero was not contained in the bootstrap 95% confidence interval for the total effect of the model (95%CI = (−0.423, −0.152)). The direct effect of the model did not include a zero value (95%CI = (−0.396, −0.123)), and the indirect effect of the model did not contain a zero value (95%CI = (−0.071, −0.007)). This model demonstrated a good fit for all of the statistical indicators (χ^2^/df, 1.551; TLI, 0.971; CFI, 0.988; RMSEA, 0.039; SRMR, 0.0246). The effect size of the mediating effect was 10.92%. The results reveal that odor identification ability had a significant mediating effect on the relationship between schizotypal traits and odor hedonic capacity as assessed by the U-Sniff test.

For the interpersonal–affective factor, which is a factor of the SPQ-C, the mediating effect was evaluated by taking the interpersonal–affective subscale score as the independent variable, the odor pleasantness score as the dependent variable, and the odor identification score as the mediating variable (Model 2, see Figure 2b). In Model 2, zero was not contained in the bootstrap 95% confidence interval for the total effect of the model (95%CI = (−0.305, −0.099)). The direct effect of the model did not include a zero value (95%CI = (−0.283, −0.080)), and the indirect effect of the model did not contain a zero value (95%CI = (−0.060, −0.005)). Model 2 is a saturated model (CFI, 1.000; RMSEA, 0.180). The effect size of the mediating effect was 12.56%. The results reveal that odor identification ability had a significant mediating effect on the relationship between the interpersonal–affective factor and odor hedonic capacity as assessed by the U-Sniff test. 

#### 3.2.2. The Mediating Effect of Odor Identification Ability on Schizotypal Traits and Odor Hedonic Capacity Assessed by the CPS-C

We treated the total SPQ-C score as the independent variable, the total CPS-C score as the dependent variable, and the odor identification score as the mediating variable (Model 3, see Figure 2c). In Model 3, zero was not contained in the bootstrap 95% confidence interval for the total effect of the model (95%CI = (−0.340, −0.002)). The direct effect of the model included the zero value (95%CI = (−0.291, 0.031)), and the indirect effect of the model did not contain a zero value (95%CI = (−0.092, −0.014)). This model demonstrated a good fit for all of the statistical indicators (χ^2^/df, 2.326; TLI, 0.928; CFI, 0.959; RMSEA, 0.061; SRMR, 0.0497). The effect size of the mediating effect was 25.99%. The results reveal that odor identification ability had a significant mediating effect on the relationship between schizotypal traits and odor hedonic capacity as assessed by the CPS-C. 

We treated the interpersonal–affective subscale score of the SPQ-C as the independent variable, the total CPS-C score as the dependent variable, and the odor identification score as the mediating variable in the model (Model 4, see Figure 2d). In Model 4, zero was contained in the bootstrap 95% confidence interval for the total effect of the model (95%CI = (−0.237, 0.027)). The direct effect of the model included the zero value (95%CI = (−0.196, 0.062)), and the indirect effect of the model did not contain a zero value (95%CI = (−0.071, −0.010)). This model demonstrated a good fit for all of the statistical indicators (χ^2^/df, 3.731; TLI, 0.883; CFI, 0.953; RMSEA, 0.088; SRMR, 0.0408). The effect size of the mediating effect was 34.62%. The results reveal that odor identification ability had a significant mediating effect on the relationship between the interpersonal–affective factor and odor hedonic capacity as assessed by the CPS-C.

### 3.3. Multi-Group Analysis

Based on chi-square testing, no significant difference was noted in children and adolescents based on gender in mediating Model 1 (Delta-DF, 5; Delta-CMIN, 2.654; *p* = 0.753), Model 2 (Delta-DF, 3; Delta-CMIN, 4.763; *p* = 0.19), and Model 3 (Delta-DF, 7; Delta-CMIN, 0.614; *p* = 0.265), but not mediating Model 4 (Delta-DF, 5; Delta-CMIN, 12.464; *p* = 0.029) (Table 2).

## 4. Discussion

In the present study, we explored the relationships among schizotypal traits, odor identification ability, and odor hedonic capacity in children and adolescents. We found that schizotypal traits were negatively correlated with both odor identification ability and odor hedonic capacity in children and adolescents after controlling for age and gender. Odor identification ability also played a mediating role in the relationship between odor hedonic capacity and schizotypal traits, especially the interpersonal–affective factor. 

The current study indicated a negative correlation between schizotypal traits and odor identification ability in children and adolescents after controlling for age and gender. This finding is consistent with the results of previous studies in adults, which demonstrated that schizophrenia patients and individuals with schizotypal traits had olfactory identification impairment [16,17]. The results also show that a deficit in odor identification ability may be a promising biomarker of schizophrenia [17]. Moreover, our results reveal that negative schizotypal traits, namely the interpersonal–affective factor, had a significantly negative correlation with odor identification. These findings are consistent with previous studies, which found that a negative correlation existed between deficits in olfactory identification ability and negative symptoms in schizophrenia patients [15,27,28,29,30,31,32] and revealed that abnormalities of orbitofrontal–limbic neurocircuitry in common pathways may be the basis for processing both olfactory and negative symptoms in schizophrenia patients [33]. A previous study has also shown that olfactory function could be a biomarker in children with neuropsychiatric disorders, including schizophrenia [34].

The current study also found that the odor hedonic capacity measured by the CPS-C and U-Sniff was negatively correlated with schizotypal traits, especially the interpersonal–affective factor, in children and adolescents after controlling for age and gender. Li et al. [14] observed that individuals with schizotypal traits presented more severe chemosensory anhedonia than normal individuals and that the negative schizotypal trait was negatively correlated with chemosensory hedonic capacity. This result is consistent with our finding in children and adolescents that odor hedonic capacity was negatively correlated with schizotypal traits, especially the interpersonal–affective factor. Several studies have found abnormalities in odor pleasantness judgment in schizophrenia patients [5,35]. Additionally, schizophrenia patients with negative symptoms also had abnormal hedonic processing of odor [36]. Studies have indicated that the deficits in odor hedonic capacity may be related to physical anhedonia in schizophrenia patients [6,37]. We suggest that this type of deficit may also develop in children and adolescents with schizotypal traits. Kamath et al.’s study [38] on youths has also suggested that hedonic capacity may be a marker of future conversion to psychosis. 

The mediating models demonstrated that odor identification ability played a mediating role in the relationship between odor hedonic capacity and schizotypal traits, especially negative schizotypal traits, in children and adolescents. The results show that negative schizotypal traits can predict odor hedonic capacity directly and indirectly, a finding that is consistent with previous studies [17,36,39,40,41,42]. A possible explanation for this is that individuals with schizotypal traits have a low odor hedonic capacity due to deficits in odor identification [43,44]. In recent years, studies have found that olfactory training may not only restore olfactory function, including odor identification ability, but also reduce depressive symptoms and improve emotional health [45,46,47,48,49]. Our findings suggest that developing olfactory identification abilities in individuals with schizotypal traits may improve their odor hedonic capacity. Additionally, improving the odor identification ability through olfactory training may make the olfactory hedonic capacity malleable, indicating the value of olfactory training for the prevention and improvement of schizotypal traits or schizophrenia spectrum disorder. 

Several limitations exist in the current study. First, the current study was a cross-sectional study; longitudinal designs are recommended to investigate the correlations between variables in more depth. Second, the study found a possible mediator in the relationship between schizotypal traits and odor hedonic capacity, and the results indicate that developing odor identification abilities through olfactory training may improve the odor hedonic capacity in individuals with schizotypal traits. Future studies could further explore the effect of olfactory training in individuals with schizotypal traits. Third, the current study only explored the role of odor identification ability. However, a previous study showed that an olfactory threshold may influence the evaluation of an odor’s pleasantness [50]. Therefore, an olfactory threshold should be considered in future studies. Fourth, the current study explored the relationship between schizotypal traits, odor identification ability, and odor hedonic capacity in children and adolescents. It is critical to investigate whether the current findings could be replicated in and extended to early onset schizophrenia patients.

## 5. Conclusions

In summary, schizotypal traits were found to be negatively correlated with both odor identification ability and odor hedonic capacity in children and adolescents, while odor identification ability was found to mediate the relationship between schizotypal traits, especially the interpersonal–affective factor, and odor hedonic capacity. Our study indicated that improving the odor identification ability through olfactory training may have a positive influence on the odor hedonic capacity in individuals with schizotypal traits. 

## Figures and Tables

**Figure 1 brainsci-12-00534-f001:**
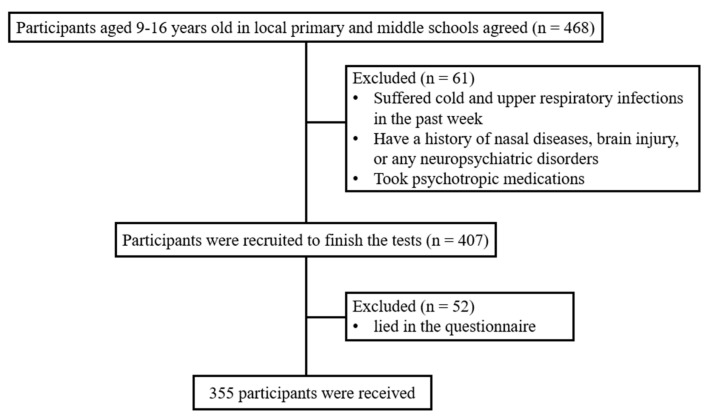
Flow diagram showing the subject recruitment process in the study.

**Figure 2 brainsci-12-00534-f002:**
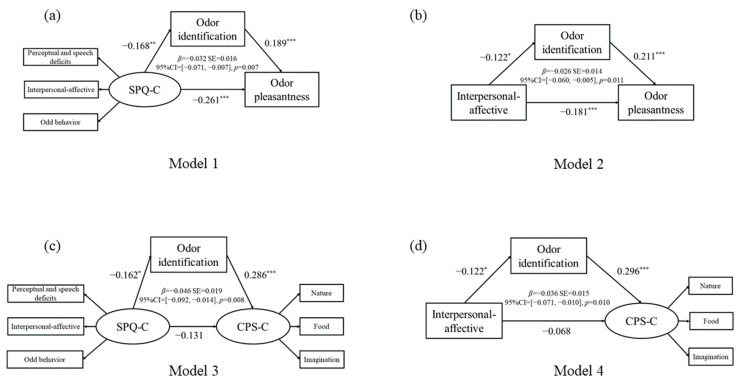
Mediating models. (**a**) Significant mediating effect in the relationship between the SPQ-C and odor pleasantness as mediated by odor identification. (**b**) Significant mediating effect in the relationship between the SPQ-C interpersonal–affective subscale and odor pleasantness as mediated by odor identification. (**c**) Significant mediating effect in the relationship between the SPQ-C and the CPS-C as mediated by odor identification. (**d**) Significant mediating effect in the relationship between the SPQ-C interpersonal–affective subscale and the CPS-C as mediated by odor identification. * *p* < 0.05, ** *p* < 0.01, *** *p* < 0.001.

**Table 1 brainsci-12-00534-t001:** Descriptive statistics and correlation analysis for the SPQ-C, odor pleasantness score, odor identification score, and the CPS-C.

Variable	M ± SD	1.	2.	3.	4.	5.	6.	7.	8.	9.
1. SPQ-C total	8.69 ± 3.76	–								
2. Perceptual and speech deficits	4.28 ± 1.93	0.845 **	–							
3. Interpersonal–affective	3.63 ± 1.95	0.807 **	0.440 **	–						
4. Odd behavior	0.78 ± 0.93	0.594 **	0.420 **	0.254 **	–					
5. Odor pleasantness score	67.41 ± 14.20	−0.256 **	−0.190 **	−0.207 **	−0.209 **	–				
6. Odor identification score	10.81 ± 1.53	−0.122 **	−0.075	−0.114 *	−0.096	0.186 **	–			
7. CPS-C total	53.62 ± 9.30	−0.174 **	−0.130 *	−0.129 *	−0.163 **	0.179 **	0.251 **	–		
8. Nature	13.92 ± 2.75	−0.217 **	−0.207 **	−0.133 *	−0.169 **	0.156 **	0.229 **	0.652 **	–	
9. Food	23.62 ± 4.71	−0.096	−0.077	−0.058	−0.106 *	0.135 *	0.254 **	0.875 **	0.441 **	–
10. Imagination	16.09 ± 4.20	−0.134 *	−0.066	−0.132 *	−0.130 *	0.144 **	0.147 **	0.805 **	0.295 **	0.526 **

Note: SPQ-C, Schizotypal Personality Questionnaire—Child; CPS-C, Chemosensory Pleasure Scale for Children. ** p* < 0.05, *** p* < 0.01.

**Table 2 brainsci-12-00534-t002:** Results of chi-square difference tests in the multi-group analysis.

Model Description	Mediating Model	DF	CMIN	*p*	NFI	IFI	RFI	TLI
					Delta-1	Delta-2	Rho-1	Rho-2
Structural weights	1	5	2.654	0.753	0.012	0.013	−0.033	−0.037
	2	3	4.763	0.19	0.113	0.113	–	–
	3	7	0.614	0.265	0.015	0.016	−0.01	−0.011
	4	5	12.464	0.029	0.049	0.051	0.018	0.02

Note: IFI, incremental fit index; NFI, normed goodness-of-fit index; RFI, relative fit; TLI, Tucker-Lewis index.

## Data Availability

The data presented in this study are available on request from the corresponding author.

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
