# Peer review of "Odor Identification Ability as a Mediator of Schizotypal Traits and Odor Hedonic Capacity in Non-Clinical Children and Adolescents"

_brainsci, 2022, doi:10.3390/brainsci12050534_

Round 1
Reviewer 1 Report
This is a study investigating the mediational role of odor identification ability of schizotypical traits and odor hedonic capacity in Children. The paper is well-written and is of interest for the readers. However, prior to its publication I would recommend several minor changes.
Anhedonia is a clinical marker of schizophrenia, as well as for depressive disorders. Both disorders should be mentioned at the beginning of the paper.
Schizophrenia may not be considered a "psychological disorder". It is a mental health disorder or a psychiatric disorder. Please, correct it.
The introduction section is brief. It should be expanded. I would propose to expand the references about the relationship between anhedonia and personality traits (schizotypical).
Inclusion and exclusion criteria should be better clarified. I would consider expanding the explanation about the selection of participants.
With regard to the data analysis, did the authors include covariates or confounding factors in the analyses?
The authors concluded that odor identification ability may be a promising biomarker of schizophrenia. It is hypothesized to be a clinical marker, or the biological (including neuroimaging) underpinnings act as a biomarker?
The conclusions section is brief. I recommend to expand them by including something about future perspectives.
Author Response
This is a study investigating the mediational role of odor identification ability of schizotypical traits and odor hedonic capacity in Children. The paper is well-written and is of interest for the readers. However, prior to its publication I would recommend several minor changes.
Anhedonia is a clinical marker of schizophrenia, as well as for depressive disorders. Both disorders should be mentioned at the beginning of the paper.
Reply: Thanks for your helpful suggestion! In the revised manuscript, we have added "Anhedonia, the declining capacity to experience pleasant emotions, is a marker of mental health disorders, including schizophrenia [1,2]and depressive disorders [3,4]" in line 39.
Schizophrenia may not be considered a "psychological disorder". It is a mental health disorder or a psychiatric disorder. Please, correct it.
Reply: Thanks for pointing out this mistake! We have corrected "psychological disorder" with "mental health disorder" (please see line 39)
The introduction section is brief. It should be expanded. I would propose to expand the references about the relationship between anhedonia and personality traits (schizotypical).
Reply: Thank you for your valuable and thoughtful comments! We have added the relationship between anhedonia and schizotypal personality traits, "Compared with healthy controls, individuals with high schizotypal traits more probably suffer anhedonia including social and physical anhedonia than those with low traits [11-13]." (see lines 47-48)
Inclusion and exclusion criteria should be better clarified. I would consider expanding the explanation about the selection of participants.
Reply: Thanks for pointing out this issue! We have clarified and added more details for the inclusion and exclusion criteria about selection of participant according to your suggestion in the revised manuscript.
"Four hundred and sixty-eight children and adolescents aged 9-16 years old agreed to participate in this study during 1st January to 31st March, 2020. All the participants were from local primary and middle schools in Guangdong, China. The exclusion criteria were as follows: (1) suffered cold and upper respiratory infections in the past week; (2) had history of nasal diseases, brain injury, or any neuropsychiatric disorders; (3) took psychotropic medications. After excluded 61 participants, 407 participants finished the tests. We then excluded 52 participants who lied in the questionnaire. Finally, 355 participants aged 9–16 (147 boys and 208 girls, mean age = 12.38 ± 3.48 years) remained for recruitment in the sample (see Figure 1). As a cross-sectional study to have a 0.95 power estimate, an effect size (0.3) and the α of 0.05 in the G*Power program were specified to calculate the necessary samples [19]. The sample size in our study is far more than the result which is proposed by the G*Power program. All the testing process were conducted in quiet and well-ventilated rooms. " (please see lines 68-80)
With regard to the data analysis, did the authors include covariates or confounding factors in the analyses?
Reply: Thank you for raising this issue! We performed partial correlation analyses to examine the influence of age and gender among these variables. "After applying partial correlation for potential confounders such as age and gender, the associations remained significant (all p < 0.05) excluding correlation between odor identification and CPS-C imagination (r = 0.08, p = 0.132)." (please see lines 122-123, 140-143)
The authors concluded that odor identification ability may be a promising biomarker of schizophrenia. It is hypothesized to be a clinical marker, or the biological (including neuroimaging) underpinnings act as a biomarker?
Reply: Thank you for your correction and comment! Several studies have revealed that the deficit of odor identification ability may be a useful biomarker for mental health disorders such as depressive disorders, schizophrenia and bipolar disorders. Accordingly, we hypothesized it is a biomarker which is acted as biological underpinnings.
The conclusions section is brief. I recommend to expand them by including something about future perspectives.
Reply: Thank you for your helpful suggestion! Schizotypal traits are multimodal entity that exist on a psychotic continuum and the risk factor for schizophrenia spectrum illness and full-blown psychosis. As a result, we presume that the relationship may be also valid in schizophrenia spectrum disorder. In the revised Conclusions, we have added "Accordingly, our results might provide a new view of improving olfactory hedonic capacity through olfactory training in the schizophrenia spectrum disorder." (please see lines 290-292)
Reviewer 2 Report
dear colleagues,
nice work.
title: mean age ~13 should be adolescents or at least children and adolescents. correct in abstract too and you may want to add work non-clinical sample
abstract onwards: present actual p-value this is important for future meta-analyses.
introduction: covered most previous literature one old/important article was overlooked and you may want to consult it https://pubmed.ncbi.nlm.nih.gov/24880865/
methods: present key elements of study design early in the paper its a cross-sectional study
identify the setting, locations, and key dates, including recruitment and data collection periods; include the eligibility criteria, as well as the sources and methods of participant selection
mention any steps to mitigate potential bias sources
explain how the sample size was determined - 355 (i presume this is slightly underpowered) + give a flow diagram of participant selection
provide 95 percent ci for results
other analyses, such as subgroup and interaction analyses, as well as sensitivity analyses, should be reported.
discuss the study's shortcomings, taking into mind potential sources of bias or imprecision.
discuss the amount and direction of any potential bias.
discuss the present data's clinical implications and utility.
Author Response
title: mean age ~13 should be adolescents or at least children and adolescents. correct in abstract too and you may want to add work non-clinical sample
Reply: Thank you for your suggestion! We have corrected “children” to “children and adolescents” in the revised manuscript and added “non-clinical” in the title and abstract. (please see lines 3 and 20).
abstract onwards: present actual p-value this is important for future meta-analyses.
Reply: Thank you for your kind suggestion! In the revised manuscript, we have presented the actual p-value in the abstract. (please see lines 24-26)
introduction: covered most previous literature one old/important article was overlooked and you may want to consult it https://pubmed.ncbi.nlm.nih.gov/24880865/
Reply: Thank you for your recommendation and addition! It is our ignorance that led to ignore this important paper. We have read this paper carefully and cited it appropriately in the revised manuscript. (please see line 47)
methods: present key elements of study design early in the paper its a cross-sectional study
Reply: Thank you for your carefulness to point this out! In the revised manuscript, we have mentioned that the present study is a cross-sectional study, and it is also a limitation in our study. Therefore, other experimental or longitudinal designs are recommended to explore more deeply the relationships between variables. (please see line 76 and 270-272)
identify the setting, locations, and key dates, including recruitment and data collection periods; include the eligibility criteria, as well as the sources and methods of participant selection
Reply: Thank you for this valuable suggestion! In our study, we have further refined the process of choosing participant and detailed procedure according to your comments and suggestions." Four hundred and sixty-eight children and adolescents aged 9-16 years old agreed to participate in this study during 1st January to 31st March, 2020. All the participants were from local primary and middle schools in Guangdong, China. The exclusion criteria were as follows: (1) suffered cold and upper respiratory infections in the past week; (2) had history of nasal diseases, brain injury, or any neuropsychiatric disorders; (3) took psychotropic medications. After excluded 61 participants, 407 participants finished the tests. We then excluded 52 participants who lied in the questionnaire. Finally, 355 participants aged 9–16 (147 boys and 208 girls, mean age = 12.38 ± 3.48 years) remained for recruitment in the sample (see Figure 1). As a cross-sectional study to have a 0.95 power estimate, an effect size (0.3) and the α of 0.05 in the G*Power program were specified to calculate the necessary samples [19]. The sample size in our study is far more than the result which is proposed by the G*Power program. All the testing process were conducted in quiet and well-ventilated rooms.". (please see lines68-80)
mention any steps to mitigate potential bias sources
Reply: We appreciate your highlighting of this issue and we have recognized gender and age in our samples as potential bias. In the revised manuscript, we have used partial correlation analysis to explore whether age and gender have a significant influence in correlation between variables mentioned in this study." After applying partial correlation for potential confounders such as age and gender, the associations remained significant (all p < 0.05) excluding correlation between odor identification and CPS-C imagination (r = 0.08, p = 0.132)." (please see lines 140-143). The results showed that even after controlling for age and gender, partial correlation analysis showed similar results.
explain how the sample size was determined - 355 (i presume this is slightly underpowered) + give a flow diagram of participant selection
Reply: Thanks for pointing it out! We have calculated the necessary samples in the G*Power program which is a statistical software. Considering having a 0.95 power estimate, it proposed the sample size is 109 which is necessary. And the number of participants contained in this study is more than 109. We have also added a flow diagram of participant selection in the revised manuscript. (please see lines 76-80 and Figure 1)
provide 95 percent ci for results
Reply: The 95 percent CI for mediating models have been performed in the study. 95 percent CI for the total effects, direct effects and indirect effects of models were reported respectively. The indirect effects of the models were also reported in the Figure 2.
other analyses, such as subgroup and interaction analyses, as well as sensitivity analyses, should be reported.
Reply: Thank you for indicating that! Multi-group analysis has been used to examine gender invariance in the mediating models. "Based on chi-square testing, no significant difference was noted with children and adolescents from gender in mediating model 1 (Delta-DF = 5, Delta-CMIN = 2.654, p = 0.753), model 2 (Delta-DF = 3, Delta-CMIN = 4.763, p = 0.19), and model 3 (Delta-DF = 7, Delta-CMIN = 0.614, p = 0.265), but not the mediating model 4 (Delta-DF = 5, Delta-CMIN = 12.464, p = 0.029) (see Table 2)." (see lines 212-219)
discuss the study's shortcomings, taking into mind potential sources of bias or imprecision.
Reply: Thank you for noticing this! We have added two more shortcomings of the study in the discussion. The first one is "The current study was a cross-sectional study, longitudinal designs are recommended to investigate the correlations between variables in more depth"(please see lines 270-272), the second one is "The current study only explored the role of odor identification ability. However, a previous study showed that olfactory threshold may also influence odor pleasantness evaluation [53]. Therefore, olfactory threshold should be considered in future study. " (please see lines 276-280)
discuss the amount and direction of any potential bias.
Reply: Thanks for your helpful suggestion! We discussed the potential influence of the potential bias such as olfactory threshold. (see lines 276-280)
discuss the present data's clinical implications and utility.
Reply: Thank you for pointing this limitation out! In the revised manuscript, we have further presented the clinical significance and potential utility among patients of schizophrenia spectrum disorder and we have added our discussion about the present study’s clinical significance. (please see lines 266-269)